 

# Identification of NPC1 as the target of U18666A, an inhibitor of lysosomal cholesterol export and Ebola infection

Feiran Lu[1], Qiren Liang[2], Lina Abi-Mosleh[1], Akash Das[1], Jef K De Brabander[2]*, Joseph L Goldstein[1]*, Michael S Brown[1]*

[1]Department of Molecular Genetics, University of Texas Southwestern Medical Center, Dallas, United States; [2]Department of Biochemistry, University of Texas Southwestern Medical Center, Dallas, United States

**Abstract** Niemann-Pick C1 (NPC1) is a lysosomal membrane protein that exports cholesterol derived from receptor-mediated uptake of LDL, and it also mediates cellular entry of Ebola virus. Cholesterol export is inhibited by nanomolar concentrations of U18666A, a cationic sterol. To identify the target of U18666A, we synthesized **U-X**, a U18666A derivative with a benzophenone that permits ultraviolet-induced crosslinking. When added to CHO cells, **U-X** crosslinked to NPC1. Crosslinking was blocked by U18666A derivatives that block cholesterol export, but not derivatives lacking blocking activity. Crosslinking was prevented by point mutation in the sterol-sensing domain (SSD) of NPC1, but not by point mutation in the N-terminal domain (NTD). These data suggest that the SSD contains a U18666A-inhibitable site required for cholesterol export distinct from the cholesterol-binding site in the NTD. Inasmuch as inhibition of Ebola requires 100-fold higher concentrations of U18666A, the high affinity U16888A-binding site is likely not required for virus entry.

*For correspondence: jef. debrabander@utsouthwestern. edu (JKDB); joe.goldstein@ utsouthwestern.edu (JLG); mike. brown@utsouthwestern.edu (MSB)

**Competing interests:** The authors declare that no competing interests exist.

## Introduction

Niemann-Pick C1 (NPC1), a lysosomal membrane protein, has emerged as the culprit in two fatal diseases. First, mutations in NPC1 underlie Niemann-Pick C disease, a devastating lysosomal storage disease in which accumulation of cholesterol in lysosomes causes abnormalities in brain, liver, lung, and other tissues, leading to death in the teenage years (*Pentchev, 2004*). Second, NPC1 is the receptor that permits cellular entry of the RNA genomes of lethal Ebola virus, Marburg virus, and other filoviruses (*Carette et al., 2011*; *Côté et al., 2011*).

In this paper, we use the term 'lysosome' in its inclusive sense to encompass late endosomes where NPC1 may also function. NPC1 mediates the egress of cholesterol that has entered lysosomes through the receptor-mediated endocytosis of low density lipoprotein (LDL) (*Liscum et al., 1989*). When cell membranes are depleted of cholesterol, LDL receptors bind circulating LDL, internalize it, and deliver it to lysosomes where acid lipase hydrolyzes the lipoprotein's cholesteryl esters (*Brown and Goldstein, 1986*). The liberated cholesterol binds to NPC2, a soluble diffusible protein in the lysosome lumen (*Sleat et al., 2004*). NPC2 transports the cholesterol to the lysosomal membrane where it transfers its cholesterol to membrane-embedded NPC1 (*Wang et al., 2010*). Designated a hydrophobic handoff, this transfer occurs in such a way that hydrophobic cholesterol is shielded from the aqueous environment (*Kwon et al., 2009*).

NPC1 is an intrinsic membrane protein of 1278 amino acids that contains 13 membrane spanning helices separated by hydrophilic loops and 19 potential N-linked glycosylation sites (*Davies and Ioannou, 2000*) (see *Figure 1*). NPC2 transfers its cholesterol to the N-terminal domain (NTD) of

**eLife digest** Cholesterol is a type of fat molecule and is a vital component of animal cell membranes. It is taken up into cells within particles called low density lipoproteins (LDLs) that are then digested in cell compartments known as lysosomes to release the cholesterol. Then, the cholesterol leaves the lysosome with the help of a transport protein called NPC1. Mutations in the gene that encodes NPC1 lead to the accumulation of cholesterol in lysosomes; this can cause a devastating illness that affects the brain, liver and other organs. The NPC1 protein also plays a crucial role in allowing Ebola viruses to infect animal cells and multiply.

U18666A is a drug that blocks the movement of cholesterol out of lysosomes and also inhibits Ebola virus infections, but it was not known what components it targets in cells. Lu et al. used a technique called ultraviolet-induced crosslinking to identify the proteins that U18666A can bind to. The experiments show that U18666A can directly bind to a site that is within a section of the NPC1 protein called the sterol-sensing domain. The binding of U18666A to this site blocks the movement of cholesterol out of lysosomes.

Lu et al.'s findings indicate that the sterol-sensing domain of NPC1 plays a crucial role in cholesterol's export from lysosomes. A future challenge is to use structural biology techniques (such as X-ray crystallography or cryo-electron microscope tomography) to understand the three-dimensional structure of NPC1.

NPC1, which is the hydrophilic sequence of 239 amino acids that projects into the lysosomal lumen after the signal peptide is cleaved (*Infante et al., 2008a*; *2008b*). NPC2 binds to the second luminal loop of NPC1, an event that may precede the cholesterol transfer to the NTD (*Deffieu and Pfeffer, 2011*). Mutations that abolish the function of either NPC2 or NPC1 lead to cholesterol accumulation in lysosomes and cause the fatal Niemann-Pick C disease (*Pentchev, 2004*; *Dixit et al., 2011*).

After its transfer to NPC1, cholesterol leaves the lysosome by a mechanism that is poorly understood. Some of the cholesterol reaches the endoplasmic reticulum (ER) where it has two actions. First, it binds to Scap, an ER protein that regulates the cleavage and activation of membrane-bound sterol regulatory element-binding proteins (SREBPs). Cholesterol binding to Scap prevents the protease-mediated release and nuclear entry of the active fragment of SREBPs, thereby blocking cholesterol synthesis and uptake from LDL (*Horton et al., 2002*). When excess cholesterol reaches the ER, it is esterified with a long chain fatty acid through the action of acyl-coenzyme A cholesterol acyltransferase (ACAT), which converts the cholesterol to its storage form as cholesteryl esters (*Brown et al., 1975*). Lysosome-derived cholesterol also reaches the plasma membrane where it fills three distinguishable pools, thereby assuring membrane integrity (*Das et al., 2014*).

The role of NPC1 in Ebola virus infection was recognized in 2011 through groundbreaking work by two groups. One group identified NPC1 through a haploid mutagenesis screen to detect genes required for Ebola infection in cultured cells (*Carette et al., 2011*). The other group identified small molecules that block Ebola infection and used ultraviolet crosslinking to trace their target to NPC1 (*Côté et al., 2011*). Mutant cells that lack NPC1 were found to resist Ebola virus infection, and infectivity was restored by transfection with a plasmid expressing full length NPC1 (*Carette et al., 2011*). In vitro binding studies demonstrated that an Ebola virus surface glycoprotein binds to luminal loop 2 of NPC1 (*Miller et al., 2012*), the same loop that binds to NPC2 (amino acids 373–620). Indeed, Ebola sensitivity was restored when NPC1-deficient Chinese Hamster Ovary (CHO) cells were transfected with a plasmid encoding only the luminal loop 2 portion of NPC1 (*Miller et al., 2012*). The latter data indicate that NPC1 serves only as the passive attachment site for the Ebola glycoprotein and that the presumed cholesterol transport activity of NPC1 is not required.

A powerful tool for the study of lysosomal cholesterol transport and Ebola virus infectivity is an androstenolone derivative termed U18666A that inhibits three enzymes in the cholesterol biosynthesis pathway (*Cenedella, 2009*). In 1989 (*Liscum and Faust, 1989*) demonstrated that U18666A has another action: it inhibits the exit of LDL-derived cholesterol from lysosomes, thereby creating the equivalent of NPC1 deficiency.

U18666A is a cationic amphiphile owing to a diethylaminoethyl chain attached to the 3-hydroxyl. Other cationic amphiphiles with very different structures also inhibit cholesterol egress from

lysosomes (*Lange et al., 2000*). The structural discrepancies created some confusion as to mechanism of inhibition. However, *Wojtanik and Liscum (2003)* reported that U18666A is more than 100-fold more potent than another cationic amphiphile, imipramine, raising the possibility that U18666A directly inhibits a protein required for lysosomal egress, whereas the low potency amphiphiles may have a nonspecific effect.

U18666A and other amphiphiles have also been reported to block the cellular entry of Ebola virus RNA (*Shoemaker et al., 2013*). However, the concentrations required are in the micromolar range rather than the nanomolar range at which U18666A blocks cholesterol egress, suggesting that U18666A acts through a nonspecific effect on Ebola virus as opposed to its specific effect on cholesterol transport.

In the current study, we used an unbiased chemical screen to identify a postulated cholesterol transport protein that is inhibited by low concentrations of U18666A. For this purpose, we synthesized **U-X**, a derivative of U18666A that contains a benzophenone moiety that attaches covalently to proteins when exposed to ultraviolet light (*Gubbens et al., 2009*). The **U-X** compound also contains an alkyne group that allows attachment of fluorophores and other functional elements through click chemistry (*Kolb et al., 2001*; *Sapkale et al., 2014*). **U-X** retains the ability to disrupt LDL-mediated cholesterol egress from lysosomes with high affinity. We also synthesized a series of U18666A derivatives that either retain or lose the ability to block lysosomal cholesterol export, and we used these as competitors to assure the specificity of the **U-X** crosslinking reaction. We identified one protein whose U18666A-binding properties matched the requirements for inhibition of cholesterol export, and that protein turned out to be NPC1.

## Results

*Figure 1* shows the predicted topology of NPC1 (*Davies and Ioannou, 2000*) together with its five known functional domains. The locations of the three loss-of-function mutations discussed in this manuscript are indicated.

*Figure 2* shows the structure of U18666A and the various derivatives that were synthesized for the current studies. For each compound, we show the inhibitory constant ($K_i$) that represents the concentration producing a 50% inhibition of the transfer of LDL-derived cholesterol from lysosomes to ER as determined with the cholesterol esterification assay (see legend to *Figure 2*). U18666A (designated **U**) is a derivative of androstenolone wherein the 3-hydroxyl is modified as a diethylaminoethyl ether, which will be charged (protonated) at neutral pH (pKa 9.41). As a control for

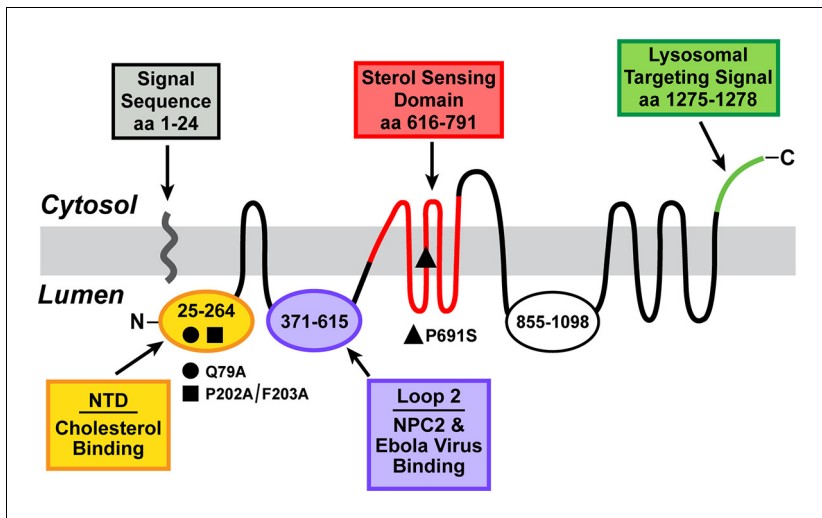

**Figure 1.** Domain structure of human Niemann-Pick C1 (NPC1). The predicted topology of the polytopic protein is based on the data of *Davies and Ioannou (2000)*. Each of the five known functional domains of the protein are shown in a different color. The locations of three loss-of-function mutations referred to in the manuscript are indicated. aa, amino acid; NTD, N-terminal domain.



**Figure 2.** Chemical structures of U18666A and derivatives used in this study. Inhibitory constant ($K_i$) values denote the concentration at which each compound inhibited the incorporation of [$^{14}$C]oleate into cholesteryl [$^{14}$C]oleate by 50% in monolayers of intact Chinese Hamster Ovary 7 (CHO-7) cells that were incubated with 10% fetal calf serum (FCS) (mean of 3–14 experiments for each compound). Assays were carried out under conditions identical to those described in *Figure 3A*.

specificity of binding, we synthesized compound **A** in which a dimethylamino group is introduced at the 17-position and the nitrogen in the ether-linked chain is replaced by a carbon. Inasmuch as compound **A** retains a dialkylamine, it will be similarly protonated at neutral pH (pKa of 10.21 as compared with 9.41 for U18666A).

In order to identify the target of U18666A, we synthesized compound **U-X** that retains the basic dialkylamino group present in U18666A, but includes a benzophenone and an alkyne group (*Figure 2*, bottom panel). When **U-X** binds to a protein and the benzophenone is exposed to 306 nM UV light, the compound attaches covalently to nearby backbone atoms of the peptide chain. Through the use of click chemistry, the alkyne group can then be attached to any compound that contains an azide group, including the fluorophore Alexa Fluor 532 Azide. The combination of UV

crosslinking and fluorescent tagging permits visualization of the protein that has bound **U-X**. *Figure 2* also shows the structures of four other compounds (**B, C, D, E**) that were used for various control experiments.

To measure the transfer of LDL-derived cholesterol from lysosomes to ER in cultured CHO cells, we use two assays: cholesterol esterification and proteolytic cleavage of SREBP-2. In both assays, we first deplete cells of cholesterol by incubating them in the absence of lipoproteins (i.e., in lipoprotein-deficient serum, LPDS) and the presence of compactin, an inhibitor of 3-hydroxy-3-methylglutaryl coenzyme A (HMG CoA) reductase, the rate-limiting enzyme in cholesterol synthesis. We add a low concentration of mevalonate, the product of HMG CoA reductase, to allow the cells to synthesize nonsterol products of the enzyme (*Goldstein and Brown, 1990*). After 16 hr, we switch the cells to a medium containing lipoproteins in the form of human LDL, fetal calf serum (FCS), or rabbit β-migrating very low density lipoprotein (β-VLDL), all of which deliver cholesterol to lysosomes via the LDL receptor (*Goldstein et al., 1983*). For the cholesterol esterification assay, we wait 3–5 hr and then add [$^{14}$C]oleate for 1–2 hr. The cells are harvested and the amount of [$^{14}$C]oleate incorporated into cholesteryl [$^{14}$C]oleate is measured by thin layer chromatography and scintillation counting (see Materials and methods). As a control for nonspecific effects, we also measure the amount of [$^{14}$C]oleate incorporated into [$^{14}$C]triglycerides. This incorporation does not depend upon lipoprotein-derived cholesterol, and it was not affected by any of the manipulations in the current experiments. The values for [$^{14}$C]triglycerides are given in the figure legends.

The second assay relies on the ability of cholesterol to block the proteolytic cleavage and nuclear localization of SREBP-2 (*Brown and Goldstein, 1997*). Synthesized as an intrinsic protein of ER membranes, SREBP-2 binds immediately to Scap, an escort protein. When the ER membranes are low in cholesterol, the Scap/SREBP-2 complex is transported to the Golgi where the SREBP-2 is cleaved by two proteases, liberating a soluble fragment that translocates to the nucleus. When LDL-derived cholesterol reaches the ER, it binds to Scap and blocks transport of the Scap/SREBP-2 complex to the Golgi. As a result, the amount of nuclear SREBP-2 declines as revealed by sodium dodecyl sulfate polyacrylamide gel electrophoresis (SDS-PAGE) and immunoblotting.

*Figure 3A* shows a cholesterol esterification assay in which 10% FCS was used as a source of LDL-cholesterol. U18666A inhibited the incorporation of [$^{14}$C]oleate into LDL-derived cholesterol with high affinity (*Figure 3A*). **U-X** retained similar inhibitory activity. On the other hand, compound **A** was nearly 100-fold less active. These experiments were repeated several times, and the calculated $K_i$ values were 0.03, 0.06, and 4.5 μM for U18666A, compound **U-X**, and compound **A**, respectively. These differences in $K_i$ values between U18666A and **A** indicate the necessity for a correctly positioned dialkylamino group to achieve inhibitory activity. In separate experiments, we showed that U18666A did not block cholesterol esterification when it is stimulated by 25-hydroxycholesterol, which enters cells without traversing lysosomes (*Abi-Mosleh et al., 2009*). We also found that U18666A does not inhibit cholesteryl [$^{14}$C]oleate formation when cholesterol is added to ER membranes in vitro.

In the SREBP-2 cleavage assay, the addition of 10% FCS led to a major reduction in the amount of nuclear SREBP-2 (*Figure 3B*). At the lowest concentration tested (0.1 μM), U18666A prevented this reduction. **U-X** also blocked at low concentrations, whereas compound **A** at 1 μM had no activity. For completeness, we show the immunoblots for the membrane-bound precursor form of SREBP-2 and NPC1, the latter used as a loading control.

To confirm that U18666A does not block the receptor-mediated uptake of LDL or the lysosomal degradation of its protein, we incubated CHO cells with $^{125}$I-labeled LDL for 6 hr, after which we measured the amount of $^{125}$I-monoiodotyrosine released into the medium. As shown in *Figure 3C*, U18666A and **U-X** did not inhibit degradation of $^{125}$I-LDL even though both compounds inhibited the lysosomal exit of cholesterol as determined by inhibition of incorporation of [$^{14}$C]oleate into cholesteryl [$^{14}$C]oleate. As a positive control, we incubated the cells with chloroquine, a cationic compound that accumulates in lysosomes where it raises the pH above the optimum for lysosomal proteases and lipases (*Goldstein et al., 1975*). Chloroquine blocked the degradation of $^{125}$I-LDL as well as the release of LDL-derived cholesterol. These data indicate that U18666A acts specifically as an inhibitor of cholesterol export and not nonspecifically as a lysosomotropic agent.

To test the hypothesis that U18666A targets NPC1, we compared its ability to block the esterification of LDL-derived cholesterol in CHO-7 cells and in TR-4139 cells, a clone of CHO-7 cells that were engineered to stably express excess NPC1 (*Figure 4*). Overexpression of NPC1 raised the

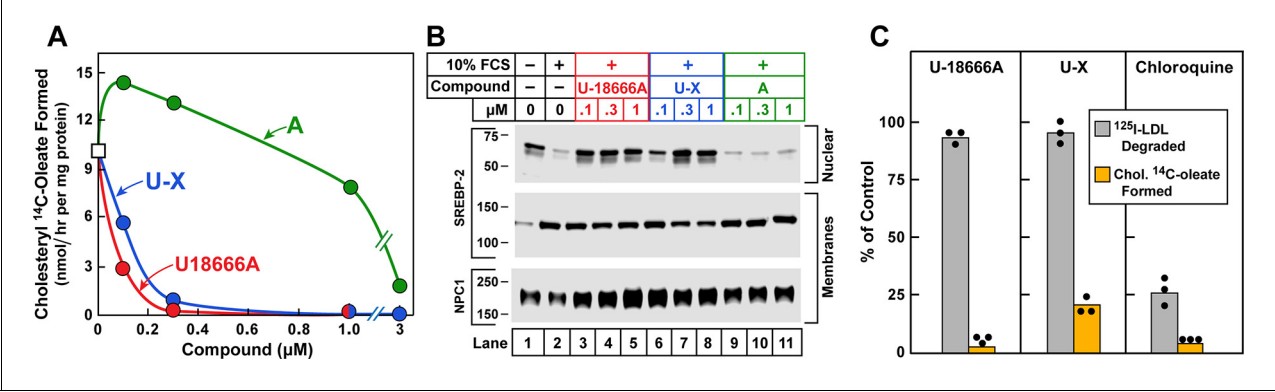

**Figure 3.** Cholesterol esterification, sterol regulatory element-binding protein (SREBP) 2 processing, and [125]I-LDL (low density lipoprotein) degradation in Chinese Hamster Ovary (CHO) 7 cells. On day 0, CHO-7 cells were set up for experiments in medium A with 5% lipoprotein-deficient serum (LPDS) at a density of 2.5 x 10[5] cells/60-mm dish. On day 2, the medium was switched to fresh medium containing 5 µM sodium compactin and 50 µM sodium mevalonate. On day 3, the medium was switched to medium B containing either 10% LPDS or 10% fetal calf serum (FCS), 50 µM compactin, 50 µM mevalonate, and various concentrations of the indicated compound and then incubated for either 7 hr (**A**) or 6 hr (**B**) as described below. (**A**) Cholesterol esterification. After a 5 hr incubation with the above medium with 10% FCS and the indicated compounds, each cell monolayer was labeled for 2 hr with 0.2 mM sodium [14C]oleate (8133 dpm/nmol). The cells were then harvested for measurement of their content of cholesteryl [14C]oleate and [14C]triglycerides. Each value is the average of duplicate incubations. Values for [14C]triglyceride content in cells treated with 1 µM of U18666A, compound **A**, and **U-X** were 117, 116, and 106 nmol/hr/ mg protein, respectively. (**B**) SREBP-2 processing. After a 5 hr incubation with the above medium containing either 10% LPDS (lane 1) or 10% FCS (lanes 2–11), each monolayer received a direct addition of 20 µg/ml of N-acetyl-leu-leu-norleucinal (A.G. Scientific, San Diego, CA). After 1 hr, cells were harvested and fractionated into a nuclear extract and 10[5]g membrane fraction (*Sakai et al., 1996*). Aliquots (30 and 10 µg protein for SREBP-2 and Niemann-Pick C1 [NPC1], respectively) were subjected to sodium dodecyl sulfate polyacrylamide gel electrophoresis (SDS-PAGE). Immunoblot analysis and image scanning were carried out with monoclonal antibodies directed against SREBP-2 or NPC1 as described in Materials and methods. (**C**) [125]I-LDL degradation. On day 3, the medium was switched to medium B containing 5% human LPDS, 20 µg protein/ml of either [125]I-LDL (for [125]I-LDL degradation) or unlabeled LDL (for cholesterol esterification), 10 µM compactin, and 50 µM mevalonate in the presence of one of the following compounds: none, 0.3 µM U18666A, 0.3 µM **U-X**, and 50 µM chloroquine. For the [125]I-LDL degradation assay, cells were incubated for 6 hr with [125]I-LDL (48 cpm/ng protein), after which the medium from each monolayer was removed and its content of [125]I-monoiodotyrosine was measured as previously described (*Goldstein, 1983*). The 100% control value for [125]I-LDL degradation in the absence of any compound (none) was 4.1 µg/6 hr/mg of protein. The cholesterol esterification assay was carried out as in **A** except that the cells were pulse-labeled with 0.1 mM [14C]oleate (6515 dpm/nmol). The 100% control value for cholesteryl [14C]oleate formed was 5.9 nmol/hr/ mg protein. The content of [14C]triglycerides in cells receiving the various compounds were not significantly different: 50, 57, 55, and 81 nmol/hr/mg protein, respectively, for no addition, U18666A, **U-X**, and chloroquine. All values are the mean of triplicate incubations with individual values shown.

threshold for U18666A inhibition by more than 100-fold ($K_i$ 0.02 µM vs. 2.7 µM). Immunoblots revealed the overexpression in the TR-4139 cells (*Figure 4*, inset). These quantitative results confirm previous nonquantitative results using Filipin staining to visualize resistance to U18666A in CHO-K1 cells that were engineered to overexpress NPC1 (*Ko et al., 2001*).

Inasmuch as **U-X** retains the ability to block cholesterol exit from lysosomes (*Figure 3*), we used this compound to identify the target of U18666A. For this purpose, we studied wild-type CHO-K1 cells and 10–3 cells, a mutant CHO-K1 cell line that lacks NPC1. The cells were incubated with **U-X** alone or in the presence of U18666A or the inactive derivative compound **A**. As described in Materials and methods, cells were irradiated with UV light after which we prepared homogenates solubilized with sodium dodecyl sulfate (SDS). We used click chemistry to attach Alexa Fluor 532 azide to the alkyne on **U-X**. The proteins were then subjected to SDS-PAGE and fluorescently tagged proteins were visualized with a fluorescence imager. Several fluorescent proteins were visualized in a UV-dependent manner (*Figure 5A*, lanes 4–13). Only one of these fluorescent bands disappeared when the cells were incubated with excess U18666A, but not with compound **A** (lanes 4–6). The apparent molecular mass of this protein was 190 kDa, which matches the apparent molecular mass of NPC1 in the same gel system. The 190-kDa band was not seen when the NPC1-deficient mutant cells were subjected to the same procedure (lanes 7–9).

To confirm the identity of the 190-kDa protein, we repeated the crosslinking experiment with CHO-K1 cells and the mutant 10–3 cells (*Figure 5B*). Again, we observed that the 190-kDa band was absent in the 10–3 cells (lane 11). It was restored when we transfected the cells with a plasmid

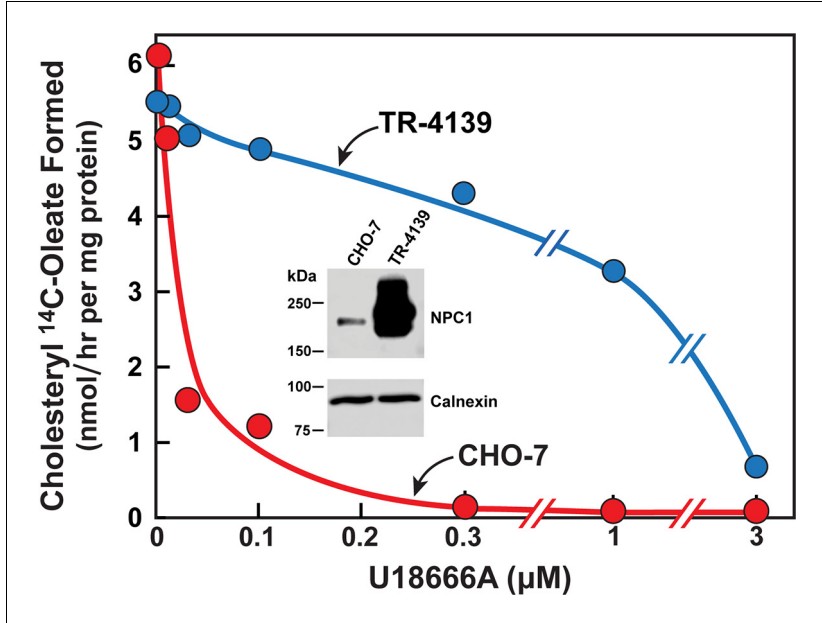

**Figure 4.** Cholesterol esterification in Chinese Hamster Ovary (CHO) 7 and TR-4139 cells that overexpress Niemann-Pick C1 (NPC1). CHO-7 cells and TR-4139 cells were set up for experiments in medium A with 5% lipoprotein-deficient serum (LPDS) as described in the legend to *Figure 3*. On day 2, the medium was switched to fresh medium containing 5 μM sodium compactin, 50 μM sodium mevalonate, and the indicated concentration of U18666A. After incubation for 4 hr, each cell monolayer was labeled for 2 hr with 0.2 mM sodium [14C]oleate (11,662 dpm/nmol). The cells were then harvested for measurement of their content of cholesteryl [14C]oleate and [14C]triglycerides. Each value is the average of duplicate incubations. The cellular content of [14C]triglycerides in CHO-7 and TR-4139 cells was 38 and 40 nmol/hr/mg protein, respectively. Inset shows immunoblots of sodium dodecyl sulfate polyacrylamide gel electrophoresis (SDS-PAGE) gels of whole cell extracts (30 μg) from the indicated cell line incubated with 0.5 μg/ml anti-NPC1 or 1 μg/ml anti-calnexin as described in Materials and methods.

encoding NPC1 but not a control protein (lanes 12 and 13). The transfected NPC1 protein migrated slightly more slowly than native NPC1, owing to the presence of epitope tags.

*Figure 6A* shows an immunoprecipitation assay to further confirm that **U-X** crosslinks to NPCl. We incubated CHO-7 cells with **U-X**, irradiated the cells with UV light, and solubilized the proteins in Nonidet P40 (NP40). We incubated the extracts with a monoclonal antibody to NPC1, and captured the antibody-bound protein on Protein A/G agarose beads. The proteins that did not adhere to the beads (designated Sup.) and the proteins eluted from the pelleted beads were then reacted with Alexa Fluor 532 using click chemistry. The proteins were subjected to SDS-PAGE and visualized with a fluorescence imager. When a control antibody was used, we observed a fluorescent doublet of proteins at about 190 kDa in the supernatant fraction. In the presence of anti-NPC1, the labeled doublet was found in the pellet fraction. The bottom panel in *Figure 6A* shows immunoblots of the same fractions with an antibody to NPC1. *Figure 6B* shows a repetition of this immunoprecipitation experiment, but in this case we incubated the cells with various derivatives of U18666A to test for specificity through competition. Crosslinking of **U-X** to NPC1 was blocked by U18666A and by compounds **B** and **C** (see *Figure 2*), all of which contain a dialkylamino group similar to U18666A and all of which block cholesterol exit from lysosomes. Crosslinking was not inhibited by compounds **A, D,** and **E**, which fail to block cholesterol exit (see *Figure 2*). Again, these data emphasize the role of a correctly positioned dialkylamino moiety appended to an adrostenolone skeleton as a requirement for achieving a specific block of cholesterol exit versus a nonspecific lysosomotropal effect of similarly basic compounds, such as **A** and **E**.

An amino acid substitution in the membrane region of NPC1 (P691S and P692S in the human and mouse proteins, respectively) allows the protein to reach its normal site in lysosomes, but prevents it from transporting cholesterol out of the organelle (*Watari et al., 1999*; *Ko et al., 2001*;

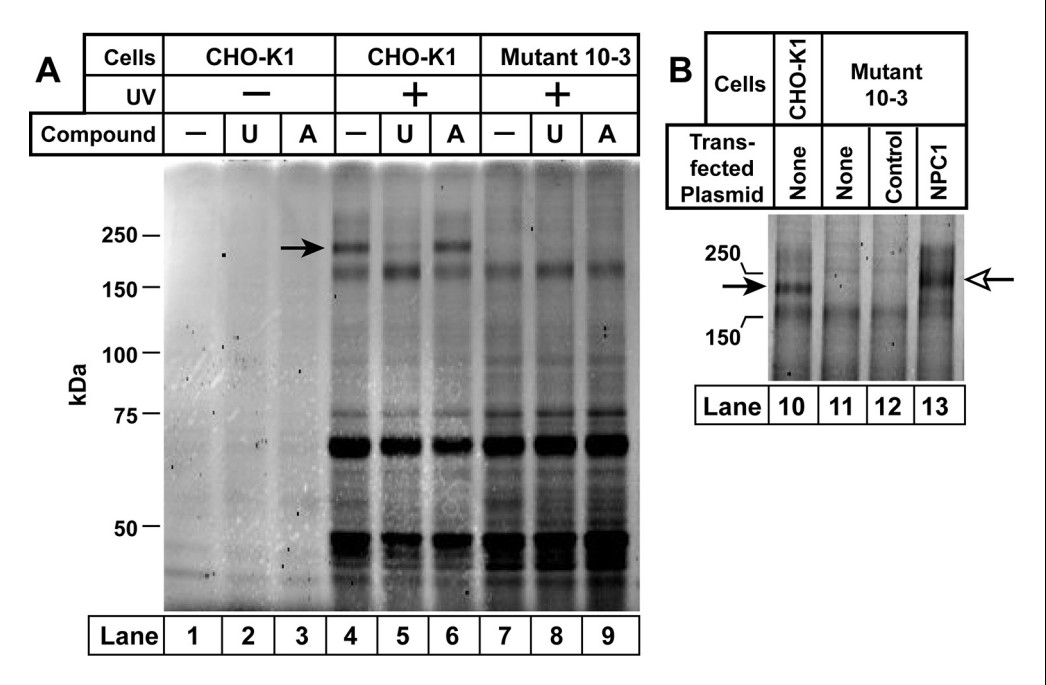

**Figure 5.** Ultraviolet (UV) crosslinking of **U-X** to Niemann-Pick C1 (NPC1) in Chinese Hamster Ovary  (CHO)-K1 cells, but not in mutant 10–3 cells. On day 0, CHO-K1 cells and 10–3 cells (mutant derivative of CHO-K1 cells that lack NPC1) were set up in a six-well plate in medium A with 5% lipoprotein-deficient serum (LPDS) (2 ml/35-mm well). (**A**) In-gel fluorescence of **U-X** binding proteins in parental CHO-K1 cells and mutant 10–3 cells that lack NPC1. On day 3, each well of cells received a direct addition of ethanol (final concentration, 0.2%) containing 0.3 μM of **U-X** crosslinker (lanes 1–9) and one of the following compounds: none (lanes 1, 4, 7); 6 μM U18666A (lanes 2, 5, 8); or 6 μM compound **A** (lanes 3, 6, 9). After incubation for 1 hr at 37°C, cells in lanes 4–9 were exposed to UV light as described in Materials and methods. Cell extracts were prepared, and the crosslinked **U-X** was fluorescently tagged using click chemistry. Proteins were then subjected to sodium dodecyl sulfate polyacrylamide gel electrophoresis (SDS-PAGE) followed by in-gel fluorescence scanning. Arrow denotes a 190-kDa protein crosslinked to **U-X** and competed by U18666A, but not compound **A**. (**B**) Fluorescent labeling of 190-kDa protein in wild-type (WT) CHO-K1 cells, but not in 10–3 cells lacking NPC1: restoration by expression of NPC1. On day 1, mutant 10–3 cells were transfected with 1 μg/well of either control plasmid (pcDNA3.1; lane 12) or plasmid encoding NPC1 (pCMV-NPC1-Flag-TEV-StrepTactin; lane 13). Nontransfected CHO-K1 and 10–3 cells were set up in parallel (lanes 10 and 11, respectively). On day 3, all cells were incubated with 0.3 μM **U-X** for 1 hr, after which they were exposed to UV light. Cell extracts were prepared, and the cross-linked **U-X** was fluorescently tagged using click chemistry, followed by SDS-PAGE and in-gel fluorescence as in Panel A. Closed arrow denotes a 190-kDa protein crosslinked to **U-X**; open arrow shows the appearance of a similar band in transfected 10–3 cells expressing epitope-tagged NPC1.

*Ohgami et al., 2004*). This mutation abolishes the binding and crosslinking of [³H]azocholesterol to NPC1 (*Ohgami et al., 2004*). To determine whether the P691S mutation affects the binding of U18666A, we transfected NPC1-deficient 10–3 cells with plasmids encoding epitope-tagged wild-type NPC1 or the P691S mutant (*Figure 7A*). The cells were incubated with **U-X** and exposed to ultraviolet (UV) light, after which the bound **U-X** was tagged with Alexa Fluor 532. After SDS-PAGE, fluorescent NPC1 was detected in cells expressing wild-type NPC1 (lane 3), which was competed by U18666A (lane 4). Fluorescent NPC1 was not detected in cells expressing the P691S mutant (lanes 5 and 6).

Another mutation that abolishes the cholesterol export function is the conversion of two adjacent amino acids in the NTD to alanine (P202A/F203A). This mutation abolishes the binding of [³H]cholesterol (*Kwon et al., 2009*), and it also reproduces the phenotype of complete NPC1 deficiency when knocked into the mouse *npc1* gene by homologous recombination (*Xie et al., 2011*). However, this mutation did not prevent crosslinking of **U-X** (*Figure 7A*, lanes 7 and 8).

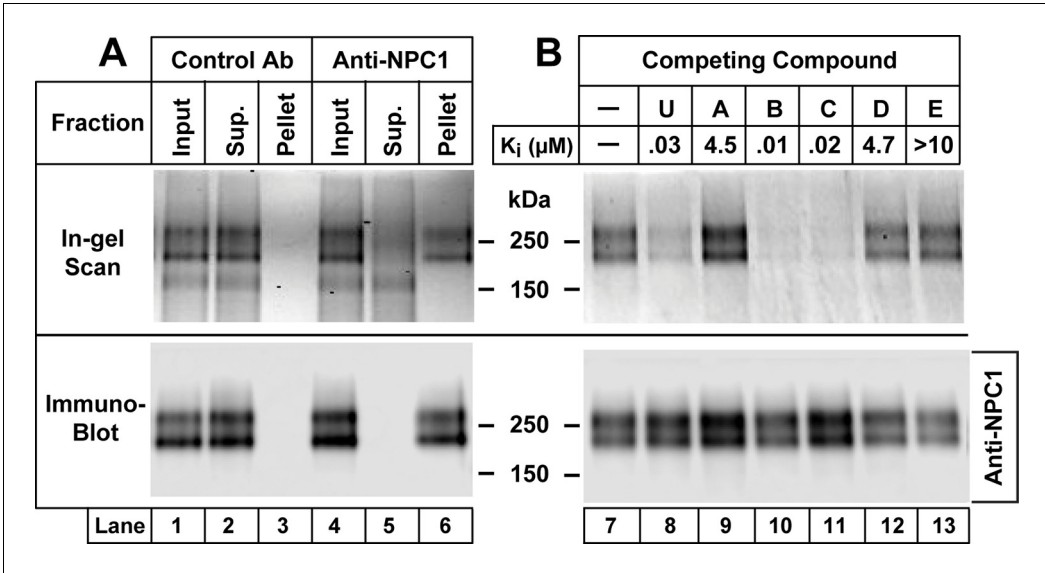

**Figure 6.** Immunoprecipitation of Niemann-Pick C1 (NPC1) crosslinked with U-X (**A**) and specificity of crosslinking reaction (**B**). On day 0, Chinese Hamster Ovary (CHO) 7 cells were set up at 1.6 x 10$^6$/150-mm dish in 25 ml medium A with 5% lipoprotein-deficient serum (LPDS) per dish as described in Materials and methods. (**A**) Immunoprecipitation with anti-NPC1 antibody. On day 3, each monolayer received 25 µl of ethanol containing U-X (final concentration, 0.3 µM). After incubation for 1 hr at 37°C, cells were exposed to ultraviolet (UV) light. Cell lysates from two dishes were pooled and incubated with 2 µg/ml control rabbit monoclonal antibody (lanes 1–3) or anti-NPC1 antibody (lanes 4–6). The solutions were applied to Protein A/G beads that were washed and eluted as described in Materials and methods. Aliquots of the input, supernatant (Sup.), and pellet fractions were obtained, and the crosslinked U-X was fluorescently tagged using click chemistry. The proteins were then subjected to sodium dodecyl sulfate polyacrylamide gel electrophoresis (SDS-PAGE) and in-gel fluorescence scanning (upper lanes) or immunoblot analysis with 0.5 µg/ml anti-NPC1 (lower lanes) as described in Materials and methods. (**B**) Specificity of crosslinking of U-X to NPC1. On day 3, each monolayer received a direct addition of 25 µl of ethanol containing U-X crosslinker (final concentration, 0.3 µM) in the absence (lane 7) or presence (lanes 8–13) of 6 µM of one of the indicated compounds whose structures are shown in *Figure 2*. After incubation for 1 hr at 37°C, cells and homogenates were processed as described in Panel A. Proteins eluted from the Protein A/G beads were subjected to fluorescent labeling using click chemistry, followed by SDS-PAGE and in-gel fluorescence scanning (upper lanes) or immunoblot analysis with 0.5 µg/ml anti-NPC1 (lower lanes) as described in Materials and methods. The K$_i$ values for the competing compounds denote the concentration at which each compound inhibited the incorporation of [$^{14}$C]oleate into cholesteryl [$^{14}$C]oleate by 50% in monolayers of CHO-7 cells that were incubated with 10% FCS (see *Figure 2*).

NPC1L1 is a close relative of NPC1 that is expressed on the surface of intestinal epithelial cells where it mediates cholesterol uptake (*Altmann et al., 2004*). *Figure 7B* shows that U-X did not crosslink to NPC1L1 when the latter was expressed in NPC1-deficient 10–3 cells (lanes 13 and 14) as compared to the wild-type NPC1 control (lanes 11 and 12). *Figure 7C* employs the cholesterol esterification assay to show that expression of wild-type NPC1 restored transport of cholesterol derived from the LDL in FCS when transfected into NPC1-deficient 10–3 cells. Transport was not restored by transfection of the P691S mutant, the P202A/F203A mutant, or NPC1L1. Immunoblots confirmed that all four proteins were expressed (*Figure 7C*, inset).

## Discussion

The current data provide further insight into the complex transport functions of NPC1, a lysosomal membrane protein required for cellular cholesterol homeostasis as well as for susceptibility to Ebola and other filoviruses. As shown in *Figure 1*, NPC1 is a polytopic membrane protein with 13 membrane-spanning helices separated by hydrophilic luminal or cytoplasmic projections. Previous studies had assigned functions to two of the hydrophilic luminal segments. The NTD, which projects into the

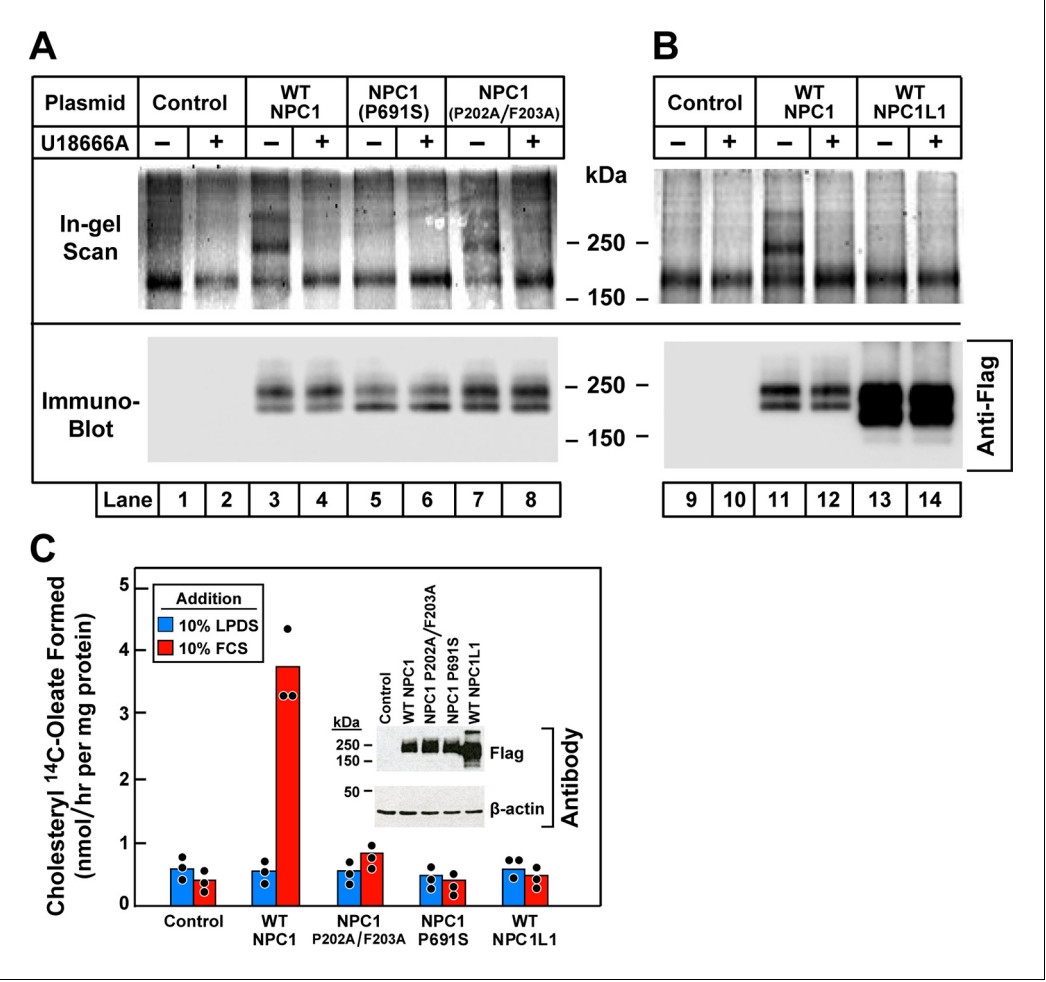

**Figure 7.** Ultraviolet (UV) crosslinking of **U-X** to transfected wild-type Niemann-Pick C1 (NPC1), mutant versions of NPC1, and wild-type NPC1L1 in 10–3 cells that lack NPC1. (**A** and **B**) Crosslinking, fluorescent labeling, and in-gel fluorescence. On day 0, 10–3 cells were set up in a six-well plate with 2 ml medium A with 5% lipoprotein-deficient serum (LPDS) per 35-mm well as described in Materials and methods. On day 1, cells were transfected by direct addition of 1 μg DNA per dish (FuGENE HD reagent) with one of the following plasmids: pcDNA3.1 control plasmid (lanes 1, 2, 9, 10); pCMV-NPC1-Flag-TEV-StrepTactin (lanes 3, 4, 11, 12); pCMV-NPC1(P691S)-Flag-TEV-StrepTactin (lanes 5, 6); pCMV-NPC1(P202A/F203A)-Flag-TEV-StrepTactin (lanes 7, 8); pCMV-NPC1L1-Flag-TEV-StrepTactin (lanes 13, 14). On day 3, all cells were incubated (without change of media) for 1 hr with 0.3 μM **U-X** crosslinker in the absence (–) or presence (+) of 6 μM U18666A, after which they were exposed to UV light. Cell extracts were prepared, and the crosslinked **U-X** was fluorescently tagged using click chemistry, followed by sodium dodecyl sulfate polyacrylamide gel electrophoresis (SDS-PAGE) and in-gel fluorescence as in *Figure 5*. (**C**) Cholesterol esterification. On day 0, 10–3 cells were set up in medium A with 5% fetal calf serum (FCS) at 2.5 x 10$^5$ cells/60-mm dish. On day 1, monolayers were washed once with Dulbecco's phosphate-buffered saline (PBS), switched to fresh medium A with 5% LPDS (devoid of penicillin and streptomycin sulfate), and then transfected with 2 μg DNA per dish with the indicated plasmids as described above. After incubation for 24 hr, cells were washed once with PBS and switched to medium A with 5% LPDS containing 10 μM sodium compactin and 50 μM sodium mevalonate. On day 3, the cells received fresh medium B containing compactin and mevalonate in the presence of either 10% LPDS or 10% FCS as indicated. After incubation for 3 hr at 37°C, each cell monolayer was pulse-labeled for 1 hr with 0.1 mM sodium [$^{14}$C]oleate (5436 dpm/nmol). The cells were then harvested for measurement of their content of cholesteryl [$^{14}$C]oleate and [$^{14}$C]triglycerides. Each value is the mean of triplicate incubations with individual values shown. The cellular content of [$^{14}$C]triglycerides in all transfected cell lines did not differ significantly in cells treated with LPDS (81–91 nmol/hr/mg) or FCS (88–105 nmol/hr/mg). Inset shows immunoblot analysis of whole cell extracts (6 μg) from the indicated transfection using a 1:1000 dilution of anti-Flag and anti-β-actin.

lumen, binds cholesterol that is delivered directly from NPC2 (*Infante et al., 2008b*; *Kwon et al., 2009*), and the second luminal loop binds NPC2 (*Deffieu and Pfeffer, 2011*) and a glycoprotein of Ebola virus that is required for release of viral RNA into the cytoplasm (*Miller et al., 2012*). Here, we find evidence for another functional domain of NPC1, namely, a domain that binds the cationic amphiphile U18666A.

At concentrations as low as 0.03 µM, U18666A inhibits the transport of LDL-derived cholesterol from lysosomes to ER as indicated by its failure to suppress SREBP-2 cleavage and its failure to undergo re-esterification by ACAT (*Figure 3*). These findings suggest a high affinity binding site for U18666A, and UV crosslinking experiments with the U18666A derivative **U-X** identified NPC1 as a protein that harbors such a binding site. U18666A derivatives that retained the ability to block cholesterol transport inhibited **U-X** crosslinking to NPC-1, whereas those that failed to inhibit transport also failed to inhibit **U-X** crosslinking (*Figure 6B*).

Although the precise location of the U18666A binding site is unknown, it does not appear to be the same as the cholesterol-binding site in the luminal NTD. We earlier showed that U18666A does not block binding of [3H]25-hydroxycholesterol to the NTD of recombinant NPC1 in vitro (*Infante et al., 2008a*). Moreover, **U-X** crosslinked normally to NPC1 bearing a P202A/F203A mutation, which abrogates [3H]cholesterol binding in vitro and cholesterol transport in intact cells (*Kwon et al., 2009*) (see *Figure 7A*).

Crosslinking of **U-X** to NPC1 was abolished by the P691S mutation, which lies in the sterol-sensing domain of NPC1 (*Figure 7A*). The sterol-sensing domain is a segment containing five transmembrane helices (helices 3–7 in NPC1) (*Nohturfft et al., 1998*; *Davies and Ioannou, 2000*). This domain was initially identified in Scap, the SREBP escort protein that is regulated by cholesterol (*Hua et al., 1996*). A similar domain is found in HMG CoA reductase, another membrane protein that is regulated by sterols (*Hua et al., 1996*). Proline 691 is a highly conserved residue that lies in the middle of the predicted third transmembrane helix of the sterol-sensing domain of NPC1. A mutation that changes this proline to serine did not prevent normal localization of NPC1 to lysosomes, but it abolished the cholesterol transport function of NPC1 (*Watari et al., 1999*; *Ko et al., 2001*) (also, see *Figure 7C*). Moreover, this mutation was also shown to prevent the crosslinking of [3H]azocholesterol to NPC1 in intact cells (*Ohgami et al., 2004*).

Considered together, the data with the P691S mutant is consistent with the idea that NPC1 contains a second cholesterol-binding site located in the sterol-sensing domain. Cholesterol initially binds to the NTD as demonstrated biochemically (*Infante et al., 2008b*) and confirmed by X-ray crystallography (*Kwon et al., 2009*). By comparing the structures of sterols bound to NPC2 and NPC1, it was possible to postulate a hydrophobic handoff by which cholesterol moves from NPC2 to NPC1 without encountering the water phase. Mutations predicted to disrupt the interaction of NPC2 with the NTD of NPC1 also disrupt cholesterol exit from lysosomes (*Kwon et al., 2009*; *Wang et al., 2010*). The hydrophobic handoff may be facilitated by the binding of NPC2 to the second luminal loop of NPC1 (*Deffieu and Pfeffer, 2011*). This binding may orient NPC2 so that it can transfer cholesterol to the NTD. The current data raise the possibility that cholesterol may move in relay fashion from the NTD to a second binding site in the sterol-sensing domain. This second site may play a role in transferring cholesterol across the lysosomal membrane. It may lack the exquisite specificity of the NTD, and it may bind U18666A and other cationic amphiphiles that thereby block cholesterol egress from lysosomes.

The postulated cholesterol-binding site in the sterol-sensing domain may require proper membrane orientation in order to bind. Although **U-X** crosslinks to NPC1 in intact cells, so far we have been unable to crosslink **U-X** to purified NPC1 in detergent solution. Moreover, our previous studies indicated that the NTD contains the only site that binds [3H]cholesterol in detergent. Thus, a single point mutation in the NTD (Q79A) abolished [3H]cholesterol binding to full length NPC1 when the protein was purified in detergents (*Infante et al., 2008a*). We are currently attempting to reconstitute purified NPC1 into phospholipid bilayers to find conditions that retain the proper membrane orientation, allowing the binding of **U-X** as well as [3H]cholesterol.

It is unlikely that binding to the sterol-sensing domain constitutes the mechanism by which U18666A inhibits Ebola virus infection. According to published data (*Shoemaker et al., 2013*; *Carette et al., 2011*; *Miller et al., 2012*), this inhibition requires concentrations of U18666A (~3–12 µM) that are several orders of magnitude higher than are required to block cholesterol transport (*Figure 3A*). Moreover, Ebola infection does not require the sterol-sensing domain of NPC1. The

second luminal loop of NPC1 is sufficient (*Miller et al., 2012*). The data raise the possibility that high concentrations of cationic amphiphiles may interact at low affinity with luminal loop 2 of NPC1, thereby blocking virus infection.

## Materials and methods

### Materials

We obtained U18666A, Dulbecco's phosphate-buffered saline (PBS), $CuSO_4$, Tris-HCl, NaCl, FCS, chloroquine, biotin, thiourea, and Benzonase Nuclease from Sigma-Aldrich, St. Louis, MO; [1-$^{14}$C] oleic acid (55 mCi/mmol) from Perkin Elmer, Waltham, MA; Zeocin and pcDNA3.1/Zeo(-) from Life Technologies, Grand Island, NY; and FuGENE HD from Promega, Madison, WI. Protein A/G Plus-Agarose Immunoprecipitation Reagent:sc 2003 from Santa Cruz Biotechnology, Dallas, TX; cOmplete, EDTA-free Protease Inhibitor Cocktail and Nonidet P40 (NP-40) from Roche Diagnostics, Indianapolis, IN; Alexa Fluor 532 Azide from Joseph M. Ready (U.T. Southwestern Medical Center); TBTA (tris[(1-benzyl-1H-1,2,3-triazol-4-yl)methyl]amine), and Diazo Biotin-Azide from Click Chemistry Tools Bioconjugate Technology Co., Scottsdale, AZ; and TCEP (tris(2-carboxyethyl)phosphine) SDS, and urea from Thermo Fisher, Rockford, IL. All tissue culture supplies and stock solutions of sodium compactin and sodium mevalonate were obtained from sources and prepared as previously described (*Das et al., 2013*). Human and newborn calf LPDS (d<1.215 g/ml), human LDL (d1.019–1.063 g/ml), and rabbit β-VLDL (<1.006 g/ml) were prepared by ultracentrifugation as described (*Goldstein et al., 1983*). Iodination of purified LDL (4 mg protein/reaction) was performed using IODO-GEN precoated tubes (Thermo Fisher Scientific Co., Grand Island, NY) (*Das et al., 2013*). More than 90% of the $^{125}$I-radioactivity in $^{125}$I-LDL was precipitable after incubation with 10% (vol/vol) trichloroacetic acid. U18666A and its derivatives (*Figure 2*) were each dissolved in ethanol and stored as a 10 mM stock solution in multiple aliquots at –20°C.

### Culture media

Medium A is a 1:1 mixture of Ham's F-12 medium and Dulbecco's modified Eagle's medium containing 2.5 mM L-glutamine (DMEM). Medium B is L-glutamine-free DMEM. All media contained 100 units/ml penicillin and 100 µg/ml streptomycin sulfate unless otherwise noted in figure legends.

### Cell culture

Stock cultures of hamster CHO-K1 cells; CHO-7 cells (a clone of CHO-K1 cells selected for growth in LPDS) (*Dahl et al., 1992*; *Metherall et al., 1989*); 10–3 cells (mutant CHO-K1 cells that lack detectable NPC1 mRNA) (*Wojtanik and Liscum, 2003*); and TR-4139 cells (a clone of CHO-7 cells stably transfected with a plasmid encoding human recombinant pCMV-NPC1-Flag-TEV-StrepTactin; see below) were maintained in monolayer culture at 37°C in a 8.8% $CO_2$ incubator.

### Plasmid constructions

pCMV-NPC1-Flag-TEV-StrepTactin encodes wild-type human NPC1 (amino acids 1–1278) followed sequentially by three tandem copies of the Flag epitope tag (DYKDDDK), one copy of the TEV cleavage site (ENLYFQ), and two copies of the StrepTactin epitope tag (WSHPQFEK) under control of the *cytomegalovirus* (CMV) promoter. This plasmid was constructed by ligating the component DNA sequences into the 5′-XbaI and 3′-HindIII sites of pcDNA3.1/Zeo(-) (*Infante et al., 2008a*). The original construct encoding human NPC1 was obtained from Origene Technologies, Rockville, MD. Point mutations (P202A/F203A and P691S) were introduced into the coding region of the NPC1 plasmid using QuikChange II XL Site-Directed Mutagenesis Kit (Agilent Technologies, Santa Clara, CA).

The coding region of each plasmid was sequenced to ensure integrity of the construct.

### Generation of TR-4139 cells expressing recombinant human NPC1

CHO-7 cells were set up on day 0 at a density of 4 x $10^5$ cells/100-mm dish in medium A with 5% (vol/vol) LPDS. On day 2, cells were transfected with 1 µg pCMV-NPC1-Flag-TEV-StrepTactin using FuGENE HD transfection reagent according to the manufacturer's instructions. Twenty-four hr after transfection, cells were switched to the above medium supplemented with 700 µg/ml Zeocin for selection. Fresh medium was added every 2–3 days until colonies formed at ~15 days. Individual

colonies were isolated with cloning cylinders, and expression of NPC1-Flag-TEV-StrepTactin was assessed by immunoblot analysis with anti-Flag antibody. Cells from single colonies were cloned by limiting dilution, maintained in medium A with 5% LPDS containing 500 μg/ml Zeocin, and hereafter referred to as TR-4139 cells.

## Cholesterol esterification assay

The rate of incorporation of [$^{14}$C]oleate into cholesteryl [$^{14}$C]esters and [$^{14}$C]triglycerides by mono-layers of CHO-K1, CHO-7, 10–3, and TR-4139 cells was measured as described previously in detail (*Goldstein et al., 1983*). In brief, 16 hr prior to experiments, cells were switched to medium containing LPDS supplemented with compactin and mevalonate. On the day of the experiment, cells were incubated for 3–5 hr with fresh medium containing one of the following lipoproteins as a source of cholesterol: FCS, LDL, or β-VLDL. Cells were then pulse-labeled for 1–2 hr with sodium [$^{14}$C]oleate-albumin complex as described in figure legends. The cells were then washed, and the lipids were extracted in hexane:isopropanol (3:2), separated on a silica gel G thin-layer chromatogram (developed in heptane:ethylether:acetic acid, 90:30:1), and quantified by scintillation counting. The amounts of cholesteryl [$^{14}$C]oleate and [$^{14}$C]triglycerides formed are expressed as nanomoles formed/hour per milligram cell protein.

## SREBP-2 processing in cultured cells

CHO-7 cells were set up for experiments as described in the legend to *Figure 3*. After incubation with the indicated compounds, nuclear and membrane fractions were prepared as described (*Sakai et al., 1996*) and then subjected to SDS-PAGE and immunoblot analysis as described below.

## UV crosslinking and fluorescent labeling with click chemistry

For fluorescent labeling of **U-X** binding proteins, CHO-K1 and 10–3 cells were set up on day 0 in 2 ml medium A with 5% LPDS at a density of 1 x $10^5$ cells/35-mm well in a six-well plate. On day 1, the cells were transfected with 1 μg of either empty plasmid or pCMV-NPC1-Flag-TEV-StrepTactin using FuGENE HD transfection reagent. On day 3, each monolayer received a direct addition of ethanol (final concentration, 0.2%) containing the indicated concentration of **U-X** crosslinker and various compounds as described in figure legends. After incubation for 1 hr at 37°C in the dark and without a change of media, the cells were irradiated for 15 min at room temperature under 306 nM UV light (Atlanta Light Bulb Co., Cat. No. G15T8E) in a UV Stratalinker 2400 apparatus (Stratagene). Cells were then washed once with PBS and resuspended in 0.1 ml buffer containing 50 mM HEPES at pH 7.8, 1.5 mM MgCl$_2$, 10 mM KCl, 100 U/ml Benzonase Nuclease, Protease Inhibitor Cocktail, and 1% (w/v) SDS. The protein concentration of each lysate was adjusted to 1.5 mg/ml using the bicinchoninic acid (BCA) protein assay. To attach a fluorophore, we employed click chemistry reactions involving copper-catalyzed azide-alkyne cycloaddition (*Kolb et al., 2001*; *Sapkale et al., 2014*). The reactions were carried out by mixing an aliquot of each SDS lysate (43 μl), in a final volume of 50 μl, with 3 μl of 1.7 mM Tris(benzyltriazolylmethyl)amine (TBTA), 2 μl of 50 mM CuSO$_4$, 1 μl of 50 mM tris(2-carboxyethyl)phosphine (TCEP), and 1 μl of 1.25 mM Alexa Fluor 532 Azide. The mixture was incubated at room temperature for 1 hr. To visualize the fluorescent-labeled proteins by in-gel fluorescence, 20-μl aliquots of each sample were mixed with 5X loading dye and subjected to 8% SDS-PAGE, after which the gels were scanned with a Typhon Imager (filter settings: excitation, 533 nM; emission, 555 nm; high sensitivity).

## Immunoprecipitation of crosslinked NPC1

For immunoprecipitation of **U-X** binding proteins, CHO-7 cells were set up on day 0 in 25 ml medium A with 5% LPDS at 1.6 x $10^6$/150-mm dish. On day 3, each monolayer received a direct addition of ethanol (final concentration, 0.2%) containing the indicated concentration of **U-X** crosslinker and various compounds as described in figure legends. After incubation for 1 hr at 37°C, the cells were subjected to UV crosslinking as described above. Cells were scraped from the dish, and the cell suspensions from two dishes were pooled. All subsequent operations were carried out at 4°C. Each pooled cell suspension was centrifuged at 1 x $10^3$ *g* for 5 min. The pellet was washed once with phosphate-buffered saline (PBS) and then solubilized with 1 ml of buffer containing 50 mM Tris-HCl at pH 7.5, 150 mM NaCl, 0.5% (v/v) NP40, and Protease Inhibitor Cocktail. Each

solubilized lysate was passed through a 22.5-gauge needle 10 times, rotated for 1 hr, and clarified by centrifugation at $1.5 \times 10^6$ $g$ for 30 min. An aliquot of the supernatant (1 mg protein) was adjusted to a final volume of 1 ml with the above buffer and precleared by rotation for 1 hr with 10 µg/ml of preimmune rabbit immunoglobulin G (IgG) and 100 µl of Protein A/G PLUS-Agarose Beads. After centrifugation at $1 \times 10^3$ $g$ for 5 min, the supernatant (designated as input) was transferred to a fresh tube and incubated with various monoclonal antibodies as described in figure legends. After rotating for 1 hr, 100 µl of Protein A/G PLUS Agarose Beads were added, followed by rotation overnight, and centrifugation at $1 \times 10^3$ $g$ for 5 min. The resulting supernatant (~1 ml) was transferred to a fresh tube and designated as supernatant. The pelleted beads were washed five times with 1 ml of wash buffer (50 mM Tris-HCl at 7.5, 150 mM NaCl, and 0.02% NP40) and suspended in 1 ml of wash buffer mixed with 1% SDS and boiled for 5 min, after which the mixture was clarified by centrifugation for 5 min at $1 \times 10^3$ $g$, and the resulting fraction was designated as pellet. Prior to click reactions, the input and supernatant fractions received a direct addition of SDS (final concentration 1%) followed by boiling for 5 min. All three fractions were then tagged with Alexa-Fluor using click chemistry as described above, after which an aliquot of each fraction representing 2% of the starting sample was subjected to SDS-PAGE, and fluorescently labeled proteins were visualized by in-gel fluorescence. After scanning the gel, the proteins were transferred to nitrocellulose, blotted with rabbit monoclonal anti-NPC1, and visualized by chemiluminescence using goat anti-rabbit IgG conjugated to horseradish peroxidase (see below).

## Immunoblot analysis

Cell fractions were subjected to electrophoresis in 1% SDS on 8% polyacrylamide gels, after which the proteins were transferred to nitrocellulose filters. The filters were incubated at room temperature with 10 µg/ml of one of the following primary antibodies as indicated in the figure legends: rabbit monoclonal IgG-22D5 against SREBP-2 (McFarlane et al., 2015); rabbit monoclonal IgG against amino acids 1261–1278 of human NPC1; (Cat. No. 134113, Abcam, Cambridge, MA); mouse monoclonal IgG against Flag epitope DDDDK (Cat. No. M185, MBL International Corp., Woburn, MA); rabbit polyclonal IgG against β-actin (Cat. No. 4970, Cell Signaling Technology, Boston, MA), and rabbit polyclonal IgG against calnexin (Cat. No. ADI-SPA-860, Enzo Life Sciences, Farmingdale, NY). Bound antibodies were visualized by chemiluminescence (SuperSignal West Pico Chemiluminescent Substrate, Thermo Scientific) after incubation with a 1:5000 dilution of either donkey anti-mouse or goat anti-rabbit IgG conjugated to horseradish peroxidase (Jackson ImmunoResearch, West Grove, PA). The images were scanned using an Odyssey FC Imager (Dual-Mode Imaging System; 2-min integration time) and analyzed using Image Studio ver. 5.0 (LI-COR, Lincoln, NE).

## Reproducibility

Similar results were obtained when each experiment was repeated on multiple occasions. Three or more independent studies were done for each experiment except for the experiment in Figure 3C, which was done twice.

## Acknowledgements

We thank our colleagues Deepak Nijhawan, Ting Han, Tongjin Zhao, Don Anderson, Joachim Seemann, and Yansong Gao for many helpful suggestions. Lisa Beatty, Shomanike Head, and Lucie Batte provided invaluable help with tissue culture. Lauren Ziemian, Chloe Benichou, and Cassandra Hamilton provided excellent technical assistance.

## Additional information

### Funding

| Funder | Grant reference number | Author |
|---|---|---|
| National Institutes of Health | HL20948 | Joseph L Goldstein<br>Michael S Brown |
| Welch Foundation | I-1422 | Jef K De Brabander |

The funders had no role in study design, data collection and interpretation, or the decision to submit the work for publication.

## Author contributions

FL, Designed research, Performed experiments, Analyzed data; QL, Synthesized compounds; LAM, AD, Performed experiments, Analyzed data; JKDB, JLG, MSB, Designed research, Analyzed data, Wrote the paper.

## Additional files

**Supplementary files**

• Supplementary file 1. Six schemes showing the synthesis of derivatives of U18666A.

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
