## [Decision Letter]

Thank you for submitting your work entitled "Identification of NPC1 as the target of U18666A, an inhibitor of lysosomal cholesterol export and Ebola infection" for consideration by *eLife*. Your article has been reviewed by three peer reviewers, one of whom is a member of our Board of Reviewing Editors, and the evaluation has been overseen by the Reviewing Editor and Michael Marletta as the Senior Editor.

The reviewers have discussed the reviews with one another and the Reviewing editor has drafted this decision to help you prepare a revised submission.

This outstanding paper uses a combination of chemical and cell biological approaches to identify NPC1 as the target of U18666A, an inhibitor of lysosomal cholesterol export and ebola virus infection. The reviewers were in agreement that the work was convincing and of broad interest and offered only relatively minor suggestions for the authors' consideration.

Major comments:

1) In previous work, the authors concluded that NPC1 has only one binding site for cholesterol, in the N-terminal domain. Others cited here had obtained data for a second site in the sterol sensing domain (SSD). The authors discuss this generously and clearly. Because of the history, it seems that the study would be greatly enhanced if the authors used mass spectrometry to identify the peptide(s) to which the crosslinker attaches. All that would be required would be IP of protein {plus minus} crosslinker, followed by tryptic digest – they would either see a change in a peptide or a loss of a peptide in the spectrum. This would strongly support binding in or near the SSD and not elsewhere in the protein. They include a SSD mutant and show lack of crosslinking but this protein may have an altered conformation, rather than a mutation in the actual binding site. Note that this experiment is not essential for publication but could be a valuable addition.

2) That U18666A does not bind N-terminal domain mutants is not a surprise (but a nice experiment) as the structure of the compound would not be predicted to bind there. Does the compound bind NPC2?

3) In Figure 3, is the U18666A acting as a pharmacological chaperone as well, increasing the amount of newly synthesized NPC1 that gets to the lysosome? Please comment.

Minor comments:

1) We suggest that the authors offer a sentence or two more in the final paragraph on the interpretation of the findings that U18666A binds NPC1, apparently in the sterol sensing domain, and blocks Ebola infection at high doses, while at the same time this region of NPC1 is not required for Ebola infection. Given these observations, it seems likely that U18666A binding to the sterol sensing domain must do something to NPC1, by changing NPC1 levels or conformation, that interferes with Ebola accessing the NPC region that it binds for infection. Providing at least one example of a possibility here in the Discussion would help the reader to reconcile these findings.

2) There are several gain-of-function mutations within the SSD of NPC1 that actually enhance cholesterol export from lysosome, such as L657F. It would be interesting to see how these gain-of-function mutations would affect the crosslinking of U18666A derivative U-X to NPC1.

---

## [Author Response]

Major comments:

*1) In previous work, the authors concluded that NPC1 has only one binding site for cholesterol, in the N-terminal domain. Others cited here had obtained data for a second site in the sterol sensing domain (SSD). The authors discuss this generously and clearly. Because of the history, it seems that the study would be greatly enhanced if the authors used mass spectrometry to identify the peptide(s) to which the crosslinker attaches. All that would be required would be IP of protein {plus minus} crosslinker, followed by tryptic digest – they would either see a change in a peptide or a loss of a peptide in the spectrum. This would strongly support binding in or near the SSD and not elsewhere in the protein. They include a SSD mutant and show lack of crosslinking but this protein may have an altered conformation, rather than a mutation in the actual binding site. Note that this experiment is not essential for publication but could be a valuable addition.*

In collaboration with a noted expert in mass spectroscopy, we tried repeatedly to identify the peptide that is crosslinked to U-X. Mass spectroscopy on hydrophobic polytopic membrane proteins is notoriously difficult. Because of the limited coverage of the hydrophobic peptides, we were unable to identify unambiguously a peptide that dropped out when crosslinked or a novel peptide that appeared in the crosslinked material.

*2) That U18666A does not bind N-terminal domain mutants is not a surprise (but a nice experiment) as the structure of the compound would not be predicted to bind there. Does the compound bind NPC2?*

In dozens of experiments with the U-X photo crosslinker, we always observed crosslinking to a band corresponding in size to NPC1. On 15% SDS gels, we never observed a crosslinked band at the molecular weight expected for NPC2 (20-25 kDa).

*3) In Figure 3, is the U18666A acting as a pharmacological chaperone as well, increasing the amount of newly synthesized NPC1 that gets to the lysosome? Please comment.*

As the reviewers note, in the immunoblot of Figure 3 we observed a slight increase in the amount of NPC1 when we added U-18666A or U-X to CHO-7 cells. This finding was not reproducible. For example, see the immunoblots in Figure 7 wherein the addition of U-18666A did not increase the amount of Wild-type NPC1.

*Minor comments: 1) We suggest that the authors offer a sentence or two more in the final paragraph on the interpretation of the findings that U18666A binds NPC1, apparently in the sterol sensing domain, and blocks Ebola infection at high doses, while at the same time this region of NPC1 is not required for Ebola infection. Given these observations, it seems likely that U18666A binding to the sterol sensing domain must do something to NPC1, by changing NPC1 levels or conformation, that interferes with Ebola accessing the NPC region that it binds for infection. Providing at least one example of a possibility here in the Discussion would help the reader to reconcile these findings.*

The reviewers make an excellent suggestion. We have added the following sentence to the last paragraph of the Discussion following the statement that Ebola virus infection requires luminal loop 2 of NPC1:

“The data raise the possibility that high concentrations of cationic amphiphiles may interact at low affinity with luminal loop 2 of NPC1, thereby blocking Ebola virus infection.”

2) There are several gain-of-function mutations within the SSD of NPC1 that actually enhance cholesterol export from lysosome, such as L657F. It would be interesting to see how these gain-of-function mutations would affect the crosslinking of U18666A derivative U-X to NPC1.

The reviewers suggest an interesting experiment that we will undertake in the future.